# Beyond the Paradigm of Weight Loss in Non-Alcoholic Fatty Liver Disease: From Pathophysiology to Novel Dietary Approaches

**DOI:** 10.3390/nu13061977

**Published:** 2021-06-08

**Authors:** Angelo Armandi, Jörn M. Schattenberg

**Affiliations:** 1Department of Medical Sciences, University of Turin, 10126 Turin, Italy; angelo.armandi@unito.it; 2Department of Internal Medicine I, University Medical Center, Johannes Gutenberg University, 55131 Mainz, Germany; 3Metabolic Liver Research Program, Department of Internal Medicine I, University Medical Center, Johannes Gutenberg University, 55131 Mainz, Germany

**Keywords:** insulin, lifestyle, non-alcoholic, steatohepatitis, fibrosis, metabolic syndrome, weight loss, time-restricted feeding, intermittent fasting, low-carb diet, liver disease

## Abstract

Current treatment recommendations for non-alcoholic fatty liver disease (NAFLD) rely heavily on lifestyle interventions. The Mediterranean diet and physical activity, aiming at weight loss, have shown good results in achieving an improvement of this liver disease. However, concerns related to compliance and food accessibility limit the feasibility of this approach, and data on the long-term effects on liver-related outcomes are lacking. Insulin resistance is a central aspect in the pathophysiology of NAFLD; therefore, interventions aiming at the improvement of insulin sensitivity may be preferable. In this literature review, we provide a comprehensive summary of the available evidence on nutritional approaches in the management of NAFLD, involving low-calorie diets, isocaloric diets, and the novel schemes of intermittent fasting. In addition, we explore the harmful role of single nutrients on liver-specific key metabolic pathways, the role of gene susceptibility and microbiota, and behavioral aspects that may impact liver disease and are often underreported in clinical setting. At present, the high variability in terms of study populations and liver-specific outcomes within nutritional studies limits the generalizability of the results and highlights the urgent need of a tailored and standardized approach, as seen in regulatory trials in Non-Alcoholic Steatohepatitis (NASH).

## 1. Introduction: Rationale for Lifestyle Interventions in NAFLD

Non-alcoholic fatty liver disease (NAFLD) represents the leading form of chronic liver disease, with a worldwide prevalence of 25% [1] and a disease burden which is projected to dramatically increase by 2030 [2], in parallel with the pandemic of metabolic-related affections, mainly obesity and type 2 diabetes (T2DM). Overall, the economic and social burden of NAFLD in Europe is high [3].

NAFLD is a multi-systemic disease which involves multi-directional metabolic derangements [4]. The term NAFLD covers different disease activities and disease stages, ranging from simple steatosis (non-alcoholic fatty liver, NAFL), to the histological evidence of lobular inflammation and ballooning (non-alcoholic steatohepatitis, NASH), which represents the progressive form of the metabolic-related liver damage that can potentially lead to cirrhosis and further complications, including hepatocellular carcinoma (HCC) [5]. Progression along this disease continuum is driven by alternating inflammatory bouts that foster excessive fibrogenesis and promote scars inside the liver parenchyma. Progression through the histological fibrosis stages to cirrhosis has been recognized as the most relevant prognostic factor in NAFLD [6]. In addition, resolution of NASH is more difficult to achieve when compared to NAFL, where no inflammatory activity is detected upon histology. Presently, liver biopsy is the reference standard to reliably grade NASH and stage fibrosis. Hence, in the absence of accepted non-invasive markers, an accurate assessment of treatment efficacy requires liver biopsy. This is frequently not performed in lifestyle intervention trials and thus results are not directly comparable with current registration studies exploring pharmacotherapy options.

Currently, no liver-directed pharmacological therapy is approved for the treatment of NAFLD, and the management relies on lifestyle interventions. The majority of individuals with NAFLD are overweight or obese and suffer from T2DM [7]. Therefore, weight loss, aiming at ameliorating the liver injury and decreasing fibrosis, is suggested by all guidelines. However, NAFLD can also occur in lean individuals, where other factors, such as an unfavorable genetic background, may lead to a comparable liver phenotype. The patients considered as lean NAFLD can develop all disease stages of NASH, supporting the concept that even in the absence of overt obesity, steatohepatitis and fibrosis can occur. [8]. In all NASH patients, a balance between harmful (metabolic) co-factors, intrinsic genetic factors, and extrinsic lifestyle choices impact on the clinical phenotype. This variability also adds to the inhomogeneity of responses observed in lifestyle interventions.

The seminal studies conducted by Vilar-Gomez et al. on 293 NAFLD individuals undergoing intensified lifestyle changes (low-fat hypocaloric diet) with psychological coaching for 52 weeks highlighted the results that nutrition and physical activity can have on liver histology: weight loss of more than 10% of the initial body weight led to a resolution of NASH in 90% of cases, and a regression of fibrosis in 45% of cases [9]. Unfortunately, only 10% of all intensively guided participants could reach this endpoint. Similar data resulted from another randomized, controlled trial conducted on 31 overweight or obese patients with biopsy-proven NASH undergoing lifestyle intervention for 48 weeks, with significant improvements in histological inflammatory activity in those who could reach at least a 7% reduction in weight loss [10]. In a routine clinical setting which offers less intensive counseling, the effects of lifestyle modifications are clearly lower, which is linked to the lower rate of adherence. 

Evidence coming from morbidly obese patients undergoing bariatric surgery have provided insightful results on the impact of rapid weight loss on liver damage. Despite differences of NAFLD in bariatric cohorts with typically lower disease activity and stages, the pronounced and sustained weight loss that occurs highlights the degree of effect that can be achieved in a short time frame. Studies conducted by Lassailly et al. showed that NASH resolves in 85% of cases at 1 year after bariatric surgery [11], and an additional beneficial effect on fibrosis can be observed across 5 years following surgery [12]. 

Histological improvements resulting from weight loss are also observed across clinical trials with investigational drugs. In a phase II trial conducted on patients with NASH, liraglutide, a synthetic agonist of glucagon-like peptide-1 (GLP-1) approved for the treatment of T2DM, showed a resolution of inflammatory activity in 39% of cases along with a weight reduction of 15% from the initial value [13]. Empagliflozin, a sodium–glucose co-transporter-2 (SGLT2) inhibitor used for the treatment of T2DM, showed a 20% resolution of liver fat content as measured by magnetic resonance spectroscopy, accompanied by a reduction in a placebo-corrected 2.5 kg of weight [14]. On the contrary, pioglitazone, a peroxisome proliferator-activated receptor (PPAR)-γ agonist, has been shown to improve NASH in up to 34% of cases with a mean weight gain of 4.7 kg [15]. 

In addition, data extrapolated by clinical trials have also demonstrated the significant influence of lifestyle interventions in modulating the course of NAFLD. Dietary recommendations and physical activity are suggested for all participants in experimental studies, in which proper, systematic designs may help to identify strong endpoints, limit heterogeneity, and predict the size of placebo responses [16]. Therefore, results emerging from placebo arms mirror the effective action of lifestyle modifications, which are strengthened by higher compliance in the setting of close clinical visits. One recent meta-analysis of the placebo groups from 39 randomized controlled trials of adults with NASH showed a significant response: up to 25% of patients given a placebo had improvements in all the histology scores, including fibrosis, with moderate heterogeneity among the different studies. Moreover, patients given the placebo had a significant reduction in steatosis, as measured by proton magnetic resonance spectroscopy, and a significant decrease in serum transaminases [17]. 

## 2. Response to Lifestyle Intervention between Genes and Environment

Diverse genetic susceptibility, as expressed by different single nucleotide polymorphisms (SNPs) in targeted genes, as well as the additional impact of metabolic co-morbidities and lifestyle habits, are together responsible for the high variability in the efficacy of lifestyle interventions. 

Genome-wide association studies (GWAS) have identified vulnerable sites in many gene loci, such as Patatin-like phospholipase domain-containing protein 3 (PNPLA3) [18], Membrane-bound O-acyltransferase domain-containing protein 7 (MBOAT7) [19] and Transmembrane 6 Superfamily Member 2 (TM6SF2) [20], which are related to a more aggressive disease phenotype, leading to a major risk for developing advanced liver disease. However, post hoc analyses conducted in a randomized controlled trial of lifestyle modifications in NAFLD populations have shown that variants in PNPLA3 were associated with better improvements in weight loss, better impacts on dyslipidemia, and greater reductions in intrahepatic fat, as evaluated by proton magnetic resonance spectroscopy [21]. These data suggest that the presence of the PNPLA3 variant may induce a better response to lifestyle intervention, and favorable results also come from a study from Sevastianova et al., where the same variant did not prevent the decrease in liver fat upon undertaking a hypocaloric low-carbohydrate diet for 6 days [22]. 

This evidence may lead to a reduced need for drug therapy in patients with PNPLA3 variants, and further studies investigating the role of other variants would potentially bring stronger results. Currently, polygenic risk scores are being evaluated, putting together different gene variants, in order to assess their potential role in preventing hard endpoints [23], and response to lifestyle interventions might be included in this novel approach. 

In addition, gene expression is continuously shaped by environmental agents, which act as epigenetic modulators of specific protein translation and synthesis. In one study conducted on mice, a high-cholesterol diet was associated with a down-regulation of key genes involved in cholesterol metabolism, such as farnesoid-X receptor (FXR), which exerts ani-inflammatory and anti-fibrotic effects upon the liver [24]. Hence, the unhealthy dietary environment may predispose the liver to unfavorable gene expression, with relevant implications in clinical settings.

The close connection between the liver and the gut supports the concept that gut microbiota (GM) correlate with the liver disease. However, these interactions are multidimensional and complex. Alterations of the gut vascular barrier and intestinal epithelial barrier are in reciprocal interaction with the GM composition and contribute to liver injury in murine models [25]. Gut integrity—a function of intestinal epithelium and dietary composition—also affects the balance between bacterial species and microbial gene richness, which may give rise to hepatic inflammation [26]. Obese people present with low microbial gene richness, and dietary intervention has been shown to improve this pattern [27]. Additionally, reductions in GM heterogeneity lead to a decrease in short-chain fatty acids and increased lipopolysaccharide (LPS)—both factors are associated with insulin resistance [28]. To add to this evidence, one longitudinal study conducted in 307 males observed a reduced risk of cardio-metabolic disease in those individuals who adhered to a Mediterranean-style dietary pattern, and this was associated with a specific GM taxonomy [29]. 

When considering the diverse response on a lifestyle treatment, ethnicity plays a relevant role. A recent meta-analysis of over two million Chinese individuals highlighted a dramatic increase in the prevalence of NAFLD, reaching 29% in the 2010s. A greater predominance of PNPLA3 variants was observed, partly explaining the 10% of NAFLD diagnosed in non-obese individuals. This might also impact on the response from lifestyle treatment. On the other hand, a major prevalence of NAFLD was observed in Westernized areas of Asia, underlying the combining impact of environmental factors [30]. The balance between genes and environment with regard to response to lifestyle treatment is still underreported in NAFLD. Randomized control trials of lifestyle interventions in Asian NAFLD populations have shown that both aerobic and resistance training, alone or in combination with dietary changes, have a positive impact on weight loss and reductions in liver fat and inflammation, however with substantial heterogeneity among studies involving different ethnicities [31]. Therefore, ethnic differences may be responsible for different outcomes for lifestyle interventions and may affect the interpretation of data.

## 3. Improving Insulin Resistance as Metabolic Endpoint for Lifestyle Intervention

Insulin resistance has been widely assessed as the major trigger of liver damage in NAFLD [32,33]. Defective action of insulin in peripheral tissues causes a reduced insulin-mediated glucose uptake in skeletal muscle, resulting in persistent hyperglycemia, and enhances lipolysis in adipose tissue, with increased levels of free fatty acids. The liver is responsible for the capture of the overflow of free fatty acids, accumulating esterified fats in lipid droplets inside the cytoplasm of hepatocytes. Insulin resistance inside the liver, linked to the alterations in post-receptor insulin signaling, leads to increased gluconeogenesis, with further harmful impacts on glycemic homeostasis. 

A mechanistic link between insulin resistance and chronic liver inflammation has been explored in studies conducted on animal models. In mice, overexpression of cytochrome P450 2E1 (CYP2E1) as a result of oxidative stress-derived liver inflammation, impairs intrahepatic insulin signaling, by decreasing the tyrosine phosphorylation of insulin receptor substrates (IRS), one key passage in the insulin metabolic pathway [34]. Hence, this interference prevents the liver from implementing a proper response to insulin. Interestingly, CYP2E1-derived inhibition of insulin signaling was partially mediated by downstream c-Jun N-terminal kinase (JNK), tightly connected to the activation and apoptosis of signal-regulating kinase 1 (ASK1). These are two major components of the hepatic inflammasome that promote apoptosis, inflammation and fibrosis, and their inhibition has been shown to improve liver damage [35], potentially acting as a target to treat insulin resistance [36]. 

Moreover, it has been shown that the crosstalk between adipose tissue and the liver, mediated by free fatty acids, results in the activation of Kuppfer cells, promoting inflammation and fibrogenesis. This pathophysiological milieu develops in the absence of obesity and T2DM, as an independent mechanism of disease [37]. One study conducted on non-obese, non-diabetic NASH individuals showed a direct association between saturated fat intake, derived indices of insulin resistance, and the postprandial rise of triglycerides, suggesting a shared pathological ground between nutrition, fats, and insulin activity, in the absence of overt metabolic-related morbidities [38]. 

Moreover, a high heterogeneity was observed in the obese population with NAFLD. Obesity is not a unique phenotype and the different compartments of adipose tissue, e.g., subcutaneous tissue versus abdominal/visceral adipose tissue, contribute differently to the disease. Visceral adipose tissue is characterized by proinflammatory activity and contributes to insulin resistance in peripheral tissues, in particular, in skeletal muscle. Muscle insulin resistance is linked to a worse metabolic phenotype. Overall, these individuals have a different clinical phenotype compared to obese patients that do not exhibit visceral adiposity and accompanying insulin-resistance. These patients are currently considered metabolically healthy obese patients. This difference may also have important implications in terms of prognosis, as well in the response to lifestyle treatment. 

High insulin levels are therefore the results of multiple drivers, involving both environmental and genetic factors, of which balance determines the phenotype and the natural history of liver disease. A diet rich in saturated fats, sucrose-enriched beverages, refined carbohydrates, high glycemic index foods, high fructose intake, high caloric foods, inserted in the picture of the Western diet, as well as harmful eating habits and sedentary lifestyles, promote hyperinsulinemia and NAFLD [39]. In particular, the detrimental effect of fructose in the context of a hypercaloric diet leads to increased de novo lipogenesis and lipotoxicity. These two major factors are involved in the progression of NAFLD to fibrosing NASH [40]. Hence, the improvement of insulin resistance would be the preferable endpoint for lifestyle interventions, with consequent beneficial effects on lipid profiles, cardio-metabolic parameters, and anthropometric measures. This complex crosstalk may be influenced by genes, as well as the co-presence of metabolic affections that would likely confer variable responses to the dietary approach.

The importance of insulin resistance in the clinical setting is supported by lifestyle interventions examining the improvement of NAFLD. Secondary or co-primary endpoints of most interventional studies are linked to the evaluation in changes of insulin resistance indices. In one randomized crossover trial conducted on obese non-diabetic biopsy-proven NAFLD individuals, lifestyle interventions using a Mediterranean diet led to a significant improvement in insulin sensitivity, as determined by a euglycemic clamp, in parallel to a reduced liver fat content, without relevant weight loss [41]. Another study conducted on adolescents with NAFLD undergoing a low-fructose diet led to a reduction in liver transaminases, accompanied by a significant improvement in systolic blood pressure and the Homeostatic Model Assessment for Insulin Resistance (HOMA-IR) [42]. The emerging role of the tight connection between liver outcomes and improvements in insulin resistance across the clinical studies suggest that insulin sensitivity should be addressed as a solid endpoint of lifestyle interventions.

## 4. Quantitative and Qualitative Aspects of Nutrition

Although association studies have highlighted a link of nutritional components, and in particular, hypercaloric diets, to liver injury in NAFLD, the specific dietary patterns or, more generally, the type of lifestyle intervention, that will reverse the disease phenotype beyond adherence to a hypocaloric diet is less clear. Resolution of steatohepatitis and regression of fibrosis are viewed as the most relevant endpoint in clinical trials. However, data on long-term outcomes, including the different incidence of metabolic concomitant affections and overall mortality, are lacking. Hence, most data are extrapolated from the efficacy of some dietary patterns in clinical trials, as well as the diverse harmful impact of different micro or macronutrients in animal models or surrogate endpoints in humans. 

Recently, the evaluation of these interconnected pathways has been explored through the approach of the Geometric Framework of Nutrition (GFN), a dimensional model that graphically integrates key aspects of nutritional systems and maps the relationship between nutrient intake and health outcomes [43]. One experimental study conducted in mice showed, with the GFN model, that a carbohydrate intake of less than 25 kj/day and protein intake of more than 10 kj/day was associated with a lower probability of developing fatty liver disease, while increased fat content was positively associated with a fatty liver [44]. This evidence suggests that not only the reduction in calories, but rather the quality and the energy content of nutritional components, can add to a successful dietary intervention. 

### 4.1. Mediterranean Diet versus Western Diet

Dietary patterns that approximate the Mediterranean diet have been repeatedly assessed in patients with metabolic diseases. In a large meta-analysis of 50 studies comprising 534,906 individuals, adherence to the Mediterranean diet was associated with a reduced risk of metabolic syndrome and, importantly, overall mortality. This was accompanied by a lower waist circumference, higher glucose tolerance, higher levels of high-density lipoprotein (HDL) cholesterol, and better levels of systolic and diastolic blood pressure [45]. 

In treating NAFLD, it is likely that easy and simple dietary patterns will be most beneficial for patients. One recent Italian randomized controlled trial, conducted by Franco et al. in 144 non-diabetic NAFLD patients, showed that a low glycemic index Mediterranean diet significantly improved hepatic steatosis, assessed by a control attenuation parameter (CAP), as well as markers of insulin resistance, as determined by HOMA-IR [46]. Interestingly, this pattern of diet ameliorates insulin resistance even in the absence of weight loss, acting on a pathophysiological level [41], and improves blood levels of transaminases (alanine aminotransferase, ALT) as surrogate markers of hepatic necroinflammatory activity, but also liver stiffness at elastography after 6 months of treatment [47].

One European prospective population study conducted on 2288 Swiss individuals without baseline hepatic steatosis for a mean time of 5.3 years has confirmed the previous evidence on a larger scale, because higher adherence to Mediterranean diet was associated with a reduced risk of developing NAFLD [48].

The Mediterranean food pattern comprises the high consumption of vegetables, fruits, mainly unrefined grains, low-fat milk, nuts; low glycemic index carbohydrates; higher proportions of monounsaturated fatty acids (MUFA) or polyunsaturated fatty acids (PUFA) with minimal saturated fats; a weekly consumption of fish, legumes, poultry and eggs; daily consumption of olive oil and a moderate consumption of red wine with meals as sources of polyphenols; and a more sporadic consumption of potatoes, red meat, sweets [49]. The model of the Mediterranean diet is based on low-glycemic index foods, which favor low levels of insulin and, more generally, a lower risk of developing insulin resistance. Therefore, the beneficial effect of the Mediterranean diet on NAFLD is likely to be dependent on the improvement of insulin sensitivity. However, any excessive intake of nutrients should be avoided, regardless of the beneficial impact on health. For instance, the anti-inflammatory benefits derived from PUFA are linked to extra-virgin olive oil, which is widely used in this dietary pattern. At the same time, an excessive intake of fatty acids (including PUFA) is associated with increased fat deposition, with harmful effects on health, highlighting the importance of balance in a dietary pattern.

Notably, the spread of the Western diet has counterbalanced the original paradigms of the Mediterranean diet, leading to less strict adherence in recent decades. In one study conducted on an Italian population, a greater use of animal proteins, processed and sugary foods, and higher intake of simple sugars and saturated fats was observed among young individuals, with respect to old subjects that continued on the original pattern of the Mediterranean diet [50].

In general, a lower intake of fibers and a higher intake of carbohydrates, saturated fats, fructose, and animal proteins favor the onset of NAFLD, in particular, in the picture of the Western diet. One prospective study conducted on 14-year-old adolescents reported a higher incidence of fatty livers at the age of 17 in those who followed a Western diet pattern, rich in take-away foods, refined cereals, and processed meats [51].

### 4.2. Behavioral Aspects That Contribute to Liver Damage

Regardless of the type of diet, behavioral aspects need to be considered. When approaching food intake, several mechanisms involve the frequency and the number of meals, and in many cases the psychological underlying background is not fully investigated. Conceiving food either as a reward, or as one way to adapt to recurrent frustrations, can result in altered nutritional behavior: craving of carbohydrates, sweet-eating, night-eating, and emotional eating are some of the labels used to identify these patterns. In particular, attitude to snacking increases intrahepatic and abdominal fat, independently from caloric contents of meals [52]. Eating before bedtime seems to be associated with a higher risk of developing NAFLD in otherwise healthy individuals [53], as does the highly perceived stressfulness [54]. Fast eating leads to an increase in total calorie intake, in particular with regard to carbohydrates, impacting on the incidence of NAFLD [52,55], and should be investigated as a potential driver of liver disease in lean individuals as well [56]. Moreover, sleep disorders are common in NAFLD population, with delayed sleep onset, poor sleep quality, and shortened sleep duration, with subsequent daily sleepiness and diminished quality of life. In one study, food intake times were switched towards the night, and was partially responsible for the sleep disturbances [57]. On the contrary, binge eating disorder was not associated with the severity of liver disease, with respect to NASH and fibrosis [58], although a higher prevalence of binge eating has been observed in NAFLD populations [59].

As for alcohol reporting, of which correct estimations in clinical setting are often challenging, dietary misbehavior is frequently underreported among obese individuals. This may be a reason of concern when approaching a failure in lifestyle intervention, possibly caused by the discrepancy between effective and reported calorie intake [60]. In recent years, mobile technology has offered novel approaches to reduce underreporting, in particular with the help of image-assisted methods that can improve the accuracy of dietary assessment [61]. With this system, food information is directly extracted from the images and the calorie content is automatically calculated through portion size estimation [62]. In one study, eating pattern among healthy individuals was monitored with the use of a mobile app. Most subjects ate frequently and irregularly for more than 14 h during the day. Regulation of eating pattern and restriction of eating duration, assisted by a novel system of data visualization (“feedogram”), resulted in reduced body weight and increased wellness [63]. These approaches might help clinicians to gather comprehensive reports of dietary patterns and habits, with the possibility of defining more precise lifestyle interventions.

## 5. Diverse Impact of Nutrients on NAFLD

Dietary patterns are highly conditioned by the geographical area, with subsequent differences in nutrient availability, but also cultural heritage, socio-economic status, and food accessibility as determined by local institutions. Moreover, the variability of the outcomes across the clinical studies, as well as the unknown effect of single nutrients on liver histology, make it hard to assess the harmful size of one specific reported dietary pattern. This aspect seems of crucial relevance in clinical practice, because it would lead to a tailored dietary pattern after addressing potentially harmful nutrients (Figure 1).

### 5.1. The Harmful Effect of Fructose Intake

For instance, in one cross-sectional study conducted on 789 individuals undergoing screening colonoscopy, the consumption of red and processed meat was associated with both NAFLD and insulin resistance [64]. Similarly, a high amount of fructose or high glycemic index foods lead to enhanced hepatic fat synthesis, while a high intake of saturated fats predisposes the liver to fat accumulation. The subsequent lipid-driven toxic damage represents the main driver of hepatic insulin resistance that arises along with NAFLD. Exogenous and endogenous advanced glycation end-products (AGEs) represent a further source of oxidative stress that might interfere with hepatic glucose metabolism in the onset of liver inflammation. Likewise, low fiber intake negatively shapes intestinal bacterial species to dysbiosis, increasing proinflammatory bacterial products that directly impact on liver metabolism. 

Fructose is among the main drivers of liver disease, because it feeds into hepatic de novo lipogenesis, thus increasing the amount of steatosis and consequent lipotoxicity, and moreover promoting insulin resistance in the context of a hypercaloric diet. One study conducted on NAFLD adolescents showed how a 6 month reduction in overall fructose intake, together with low glycemic index diet, improved both metabolic parameters (systolic blood pressure and HOMA-IR) and liver biochemistry (ALT) [42]. However, specific patterns of fructose intake have been investigated; not fructose ingestion per se, but rather its addition in sweet beverages, which has been attributed to metabolic derangements. Indeed, one recent randomized, double blind, placebo-controlled trial evaluated the impact of beverage consumption: containing fructose, sucrose (glucose-fructose disaccharide) or glucose. Interestingly, fructose- and sucrose-, but not glucose-containing beverages increased the hepatic synthesis of fatty acids, even in basal state and at a stable average total energy intake. These findings support the hypothesis of long-term fructose-induced intrahepatic metabolic changes, leading to adaptive pathways and increased basal lipogenic activity. During the 7 weeks of intervention, the authors did not detect any differences in metabolic outcomes (fasting plasma triglycerides, glucose and insulin concentrations, HOMA-IR modifications, arterial hypertension rates), concluding that chronic exposure to fructose-containing beverages impacts only on liver lipogenesis, the first hallmark in the onset of metabolic liver disease [65].

### 5.2. Exploration of High-Glycemic Index Nutrients

The detrimental effect of high glycemic index nutrients on NAFLD has been shown in multiple studies. High glycemic index foods (potatoes, white rice, white bread, honey) rapidly increase insulin blood levels, with a subsequent harsh decrease in blood glucose levels. This metabolic pattern is linked to two main aspects: the hyperinsulinemia involved in the long-term deleterious adaptation in insulin-sensitive tissues, and the quick onset of starvation caused by hypoglycemia, causing an increase in caloric intake and potentially leading to eating disorders. This eating pattern leads to hepatic fat accumulation and higher glycogen stores [66]. As with fructose consumption, a high glycemic index food pattern acts on an early step of the liver disease, without impacts on other metabolic parameters (namely, blood glucose levels, triglycerides, high-density lipoprotein cholesterol) [67]. However, longitudinal studies are needed to assess the long-term impacts of this eating pattern on the multiple components of the metabolic syndrome, as well as on the evolution of NAFLD.

In a diabetic population, high glycemic index food patterns are particularly unfavorable, and a long-term low glycemic index diet has been shown to improve glycated hemoglobin and fasting blood glucose, with respect to controls [68]. This nutritional strategy may therefore be tailored to diabetic NAFLD patients, in order to improve overall metabolic disruption.

### 5.3. AGEs and Oxidative Stress

Oxidative stress, induced by reactive oxygen species (ROS), is part of the pathophysiology in NAFLD [69]. Overproduction of ROS occurs when the mitochondrial oxidative capacity is exhausted by an excess of substrates, including saturated fats undergoing β-oxidation. This leads to impaired mitochondrial phosphorylation, which is associated with reduced ATP synthesis and caspase-mediated apoptosis. Excessive ROS can also cause lipid peroxidation, which is a strong driver of lipid-derived liver inflammation.

The susceptibility to ROS-driven hepatic oxidative stress is also influenced by the reduced availability of antioxidant molecules, such as glutathione (GSH). GSH is an ROS scavenger that enables activity of the key antioxidant enzyme GSH peroxidase. In mice fed with a high fat diet, a reduction in GSH and GSH peroxidase was detected, with reduced ability to control the inflammation and mitochondrial-derived oxidative stress, perpetuating the damage. Diet-induced weight loss reduced hepatic fat content and restored the regular transcription and synthesis of antioxidant enzymes [70]. Similar results have been observed in NAFLD patients undergoing bariatric surgery. One year after the intervention, significant differences in plasma and liver markers of oxidative stress were detected, in comparison with baseline evaluations [71].

In addition, one source of oxidative stress is AGEs. These are a biologically active group of molecules that are formed through no-enzymatic reactions between reducing sugars and proteins, lipids and nucleic acids [72]. Their synthesis greatly depends on the concentration of reactants and the half time of the proteins involved; the longer the time, the more prone to develop AGEs. Low rates of AGE formation are normal in metabolism in healthy individuals, but increased production occurs in the presence of carbohydrate overload. Hence, in the context of T2DM, these products are increasingly synthesized. In addition, a Western dietary pattern, in particular with high consumption of red meat and refined grains, sustains the onset of pro-inflammatory activity [73] and hence the development of oxidative stress, which favors AGE formation. In addition, AGEs can be synthesized at an exogenous level, in processed foods (heat treatment, caramel production, bread baking) which are prevalent in Western diets [74].

Accumulation of AGEs results in the activation of pro-inflammatory and pro-fibrotic pathways. In fact, the liver is responsible for the clearance of AGEs, through the interaction between specific receptors (RAGEs) expressed by Kuppfer cells and endothelial cells [75]. This link activates intracellular signaling involved in oxidative stress, thus perpetuating the damage.

Interestingly, AGEs have been shown to potentially discriminate between healthy individuals and NAFLD [76], as well as to distinguish between minimal steatosis versus moderate steatosis; therefore, they may likely be non-invasive biomarkers in high-risk populations [77]. In one randomized, placebo-controlled trial, the simple restriction of oral AGE intake was shown to ameliorate insulin resistance in obese individuals, suggesting that this could be a valuable nutritional intervention acting on a pathophysiological level [78].

### 5.4. Contribution of Lipids to Liver Damage

The exploration of the impact of lipid profiles in NAFLD has provided notable results. As discussed above, carbohydrates mainly act by increasing hepatic de novo lipogenesis, whereas the excessive intake of saturated fats increases intrahepatic triglyceride content, as compared to unsaturated fats, and increases the rate of lipolysis. Moreover, saturated fats seem to have the highest impact on insulin resistance and stimulate the synthesis of ceramides, which are mainly involved in the process of lipotoxicity and oxidative stress [79]. Similarly, diacylglycerols (DAGs) are intermediates of dietary fat oxidation directly implied in the disrupted hepatic glucose metabolism. By interfering with downstream regulating factors of the insulin pathway, DAGs promote insulin resistance and lipid-mediated hepatocellular damage [80].

In patients with NASH, long-term supplementation of PUFA was shown to decrease blood triglyceride levels and to improve or stabilize intrahepatic inflammatory activity [81], as well as to increase circulating levels of adiponectin, which is an adipose tissue-derived anti-inflammatory hormone [82]. The effect of non-saturated fats is mostly evident in the setting of the Mediterranean diet; nonetheless, one study conducted on an Asian Indian population showed an improvement in the grading of fatty livers and HOMA-IR in NAFLD patients consuming MUFA from vegetable oil, regardless of the overall alimentary pattern [83]. In addition, fish oil, which has been widely used as a source of unsaturated fats for the modulation of dyslipidemia and inflammation, has shown benefits in the setting of liver disease. In a randomized, double blind, placebo-controlled trial of patients with NAFLD and dyslipidemia, fish oil intake was associated with a significant improvement in glucose, cholesterol and triglyceride levels, as well the normalization of transaminases and increases in adiponectin levels. Moreover, a further beneficial effect on inflammation was reported from reductions in prostaglandins and tumor necrosis factor-α (TNF) [84].

### 5.5. Current Evidence on the Role of Fiber Supplementation

One further aspect that needs to be highlighted is the contribution of fiber intake to metabolic equilibrium. Fibers modulate gut microbiota, which is a substrate for producing short-chain fatty acids (acetate, propionate and butyrate), improving intestinal motility and enterocyte function [85]. Moreover, their positive effects on satiety lead to a better control of body weight [86]. The Western diet is characterized by a lower intake of fibers, and the subsequent reduced modulation of gut microbiota may lead to altered metabolic host status and impaired regulating pathways of enteric hormones, favoring the onset of obesity and T2DM [87]. Despite these two conditions being the key metabolic factors inducing NAFLD, evidence of the direct impact of fibers of NAFLD are lacking, due to the low level of scientific evidence [88]. Fiber supplementation seems to improve NAFLD-related surrogate outcomes, namely, Body Mass Index (BMI), HOMA-IR and transaminases [89], although prospective studies are needed to evaluate the long-term impact of fibers on hard NAFLD outcomes.

### 5.6. Alcohol and NAFLD: What Are the Proper Recommendations?

Alcohol consumption is a co-factor that exerts an injurious action on liver health. A small intake of red wine with meals (with a threshold of 30 g/day for men and 20 g/day for women) has been considered safe, with regard to substantial harm for chronic liver disease. Additionally, the regular intake of red wine is promoted, due to its beneficial effect related to the antioxidant action of resveratrol. However, when considering individuals presenting NAFLD, the beneficial action of red wine becomes challenging. Cross-sectional studies have suggested that modest consumption of alcohol is associated with a reduced degree of severity (comprising histological necroinflammation and fibrosis), with respect to abstinence [90,91]. However, in the complex metabolic scenario in which NAFLD is imbricated, the compound effect of the concomitant conditions has to be considered, because some features of alcohol metabolism (e.g., oxidative stress, hypertriglyceridemia) may have impacts in a considerable and unique manner. One large prospective population study involving 6732 individuals with metabolic syndrome and different grades of alcohol consumption showed that even low amounts of alcohol independently predict a severe phenotype of NAFLD [92]. Currently, less than moderate levels of alcohol, defined as 210 g/week for men and 140 g/week for women, are recommended most frequently, with absolute abstinence in patients with cirrhosis.

## 6. Alternative Dietary Approaches: Which Best Strategy?

The Mediterranean diet, despite its comprehensively beneficial action on NAFLD and metabolic syndrome, has several limitations, due to geographical and ethnical disparities throughout the globe. Therefore, different approaches have been evaluated in order to assess the benefit of a specific dietary pattern on the glycometabolic profile, regardless of the actions of single nutrients (Table 1).

Reducing overall daily calories leads to weight loss due to the negative energetic balance, strengthened by adding aerobic physical activity. In one prospective study conducted on obese individuals, a low-fat restricted-calorie diet, Mediterranean restricted-calorie diet, and low-carbohydrate non-restricted diet were randomly assigned for 2 years. The impact of a specific dietary pattern on weight loss varied according to gender, being higher for the low-carbohydrate group in males (mean reduction of 4.9 kg in men versus 2.4 kg in women) and for the Mediterranean diet group in women (mean reduction of 4.0 kg in males versus 6.2 kg in women). Low-fat diet resulted in smaller changes (3.4 kg in males and 0.1 kg in women). The low-carbohydrate diet showed the best rate of reduction in total cholesterol/HDL ratio (20% versus 12% in the low-fat arm), whereas the best impact on the glucose and insulin levels, in particular in the subgroup of diabetic individuals, was observed in the Mediterranean diet group [102]. These results suggest that a reduction in carbohydrates may be the preferable strategy in dietary patterns. In addition, the different impact on metabolic parameters emphasizes the importance of a tailored approach.

Therefore, the common strategy, given the impact of weight loss in ameliorating liver histology, has been so far the hypocaloric diet (approximately 1000 kcal/day or less). In particular, the model of the very low calorie diet (800 kcal/day), of which benefits have been well-assessed in diabetic populations, was applied in 45 patients with NAFLD, and comprised 19.4% fat, 43.4% carbohydrate and 33.7% protein. Overall, 34% of participants achieved a 10% weight loss, while 51% of participants achieved a more than 7% weight reduction. Notably, this dietary pattern was well tolerated and provided long-term benefits after interruption of the study. Concomitant significant improvement of HOMA-IR, transaminases and liver stiffness were observed [98].

Reducing carbohydrates results in significant improvements in intrahepatic triglycerides, hepatic insulin sensitivity and glucose production, when compared to an approach based on reducing fats, despite long-term evaluations not showing differences between the two approaches [103]. It would seem that mere calorie restriction helps in achieving the endpoint, regardless of the nutrient composition. In fact, one randomized controlled trial conducted on 60 overweight or obese NAFLD patients showed the impact of a food pattern with low saturated fat and low dairy products (called DASH, Dietary Approaches to Stop Hypertension, with less than 1000 kcal/day). After 8 weeks of intervention, a greater reduction in weight loss was observed in the DASH group as compared to the control arm (3.8 kg versus 2.3 kg), and BMI (−1.3 points versus −0.8 points). A significant reduction in HOMA-IR was observed in the DASH group (−0.8 points versus −0.2 points in the control arm). Accordingly, serum triglycerides, total cholesterol/HDL ratio, transaminases and markers of oxidative stress improved significantly in the interventional arm. In this study, the proportion of nutrients was almost the same (55% carbohydrates, 15% proteins, 30% fats), with sole differences in fat compositions [97].

Alternatively, isocaloric approaches have been proposed, aiming at modifications in the proportion of nutrients, but keeping average daily calories. Increasing protein intake (30% of the total), either animal- or plant-derived, has been shown to reduce liver fat independently of body weight, and to improve insulin resistance [100]. The relative reduction in carbohydrate intake might also have contributed to the outcome. Moreover, a reduction in fat intake inside the isocaloric dietary pattern has been shown to positively impact on liver fat, as evaluated by proton magnetic resonance spectroscopy: limiting fat intake to less than 16% of the total daily calories led to a 20% reduction in liver fat [104], and similar results were achieved when combining low saturated fats with low glycemic index foods [101]. Furthermore, one 12 week interventional study compared the impacts of a low-fat diet or Mediterranean diet on hepatic steatosis, and showed no differences between the two strategies in terms of benefits, with stronger adherence among the latter [94]. However, one isocaloric low-carbohydrate approach with a relative increase in protein intake seemed to provide the same results, with further evidence of the improvement in beta-oxidation and de novo lipogenesis [99].

Overall, two assumptions could be made. Firstly, there seems to be no clear evidence of preferring one nutrient reduction in respect to others, and the heterogeneity of the studies does not allow for reliable conclusions to be drawn. Moreover, compound modifications in nutrient proportions show that the endpoint is rather achieved with a varied and balanced diet. One large, prospective, interventional study showed, with proton magnetic resonance spectroscopy, that a resolution of hepatic fat could be obtained after 12 months of a dietary approach based on a variety of foods, with emphasis on fruits and vegetables, and with moderate carbohydrate intake, low fats, low glycemic index foods and appropriate portions of meals [93].

Secondly, the short durations of the studies do not provide strong evidence for the impact on metabolic health and liver outcomes. This last point is of crucial relevance when approaching clinical management of the disease. Long-term adherence to a dietary approach may be difficult to achieve, and solid outcomes need to be addressed. One large multicenter study conducted in the United States (the Look Action for Health in Diabetes (AHEAD) Study) enrolled diabetic overweight or obese patients with T2DM and randomly assigned them to either an intensive lifestyle intervention aiming at weight loss (with both decreased caloric intake and increased physical activity), or to diabetes support and education. The primary endpoint was death from cardiovascular causes and incidence of cardiovascular events for a maximum follow-up of 13.5 years. The study was interrupted early at 10 years for futility: no difference between the two groups in the primary outcome occurrence was observed [105]. Again, this evidence highlights the still-unmet need for clear outcomes in dietary patterns, associated with specific, disease-related hard endpoints.

## 7. Intermittent Fasting to Improve Metabolic Health

Metabolic modifications with regard to lipid and glycemic profiles can also be achieved with other manipulations, in particular by acting on the mealtime, rather than on its composition. Protracted fasting, in particular, causes deep modifications in metabolic pathways that persist over the refeeding time and thus may be one valuable alternative for shaping energy intakes. With this approach, energy restriction is not required to obtain metabolic improvement, nor is weight loss regarded as a key endpoint [106].

This approach is defined as intermittent fasting (IF), and covers different strategies, including alternate-day fasting, the 5:2 diet, or the fast-mimicking diet. In particular, the model of time-restricted feeding (TRF) [107] looks attractive for the liver and metabolic health prospective. TRF allows food intake within a definite interval of hours, following hormone circadian rhythms. The concept of “chrononutrition” has progressively gained attention, because daily metabolic rhythms were found to be dictated by molecular “clocks” acting on specific subsets of genes [108]. This complex metabolic crosstalk is hierarchically overseen by the hypothalamus, in response to exogenous and endogenous stimuli (the sleep–wake cycle according to light–dark courses as a paradigm). Peripheral organs, such as the liver, pancreas, skeletal muscle and adipose tissue, the key drivers of the systemic metabolic homeostasis, allow the perfect coordination (a true “synchronization” ) between the environment and the biochemical processes. Glucose homeostasis and insulin sensitivity are examples of time-related regulation; mealtimes, in return, regulate adiposity and body weight [109,110].

Animal studies conducted in rodents experiencing TRF have been shown to attenuate the harm of obesogenic diets [111], proportional to the fasting duration, reversing or stabilizing preexisting obesity, insulin resistance [112], as well as improving hepatic steatosis [113] and inflammatory markers [114]. Furthermore, TRF seems to have a modulating effect on GM, contributing to cyclical changes and diversity that impact on host metabolism [115]. Moreover, a long-term protective impact of TRF, related to the underlying gene regulation that provides durable effects, was detected even when it was temporarily interrupted in favor of unlimited access to food [116].

In humans, biological clocks suggest that metabolism is optimized for food intake in the morning, with insulin sensitivity and beta-cell responsiveness being higher in the morning than in the afternoon or evening. Individuals with T2DM who follow this pattern show a significant reduction in postprandial hyperglycemia [117], whereas insulin-resistant individuals show significant improvements in insulin sensitivity and, interestingly, a reduction in hunger [117]. In addition, overweight or obese individuals are more prone to weight loss following this alimentary timing [118], and a better modulation of adipokines (increase in adiponectin and decrease in leptin) has been reported [119]. Sutton et al. performed one elegant proof-of-concept study to show the benefits of an early TRF (allowing food intake for 6 h before 3 p.m.). This was one 5-week, randomized, crossover, isocaloric end eucaloric controlled feeding study conducted on overweight and pre-diabetes individuals. TRF improved insulin levels, insulin sensitivity, beta-cell responsiveness, blood pressure and oxidative stress, even in the absence of weight loss [120]. In particular, the improvement in blood pressure, even though participants had mean values in the pre-hypertensive range, is of great interest in terms of endpoints, as well as the smaller importance given to weight loss in the amelioration of metabolic health. Given the central role of insulin resistance in the pathogenesis of NAFLD, the TRF approach might provide interesting results.

## 8. Outlook and Summary—Open Issues in the Clinical Setting

The wide heterogeneity of lifestyle intervention approaches, together with the diverse endpoints assessed by the clinical studies and the high variability in study populations and duration of treatment, have considerable impacts in clinical practice.

In addition, a uniform effect to specific lifestyle intervention can hardly be achieved. One large study evaluated changes in glucose levels in 800 overweight and obese non-diabetic individuals following identical meal intakes. Here, a high variability in post-prandial glucose levels was observed, highlighting that individual recommendations might be required [121].

More than one approach seems beneficial in NAFLD populations, but a lack of standardization prevents clinicians from making reliable decisions, in particular with regard to the short-term assessment of either compliance or success of the chosen intervention. Weight loss would still be the primary endpoints of “bedside” recommendations, but more appropriate, tailored outcomes are required. In particular, in the absence of long-term hard endpoints in NAFLD populations, validation of simple tools may help in handling successes or failures in clinical settings. Reductions in CAP after 12 weeks of treatment is one example explored in the above-mentioned studies [46], as was the remission of steatosis through proton magnetic resonance spectroscopy after 12 months of intervention [93]. Alternatively, surrogate measures of insulin resistance, such as short-term improvements in HOMA-IR, might be used to assess the benefit of the intervention, even though the size effect may vary according to the chosen approach [97,103].

In fact, the assessment of one endpoint seems critically associated with the interventional strategy. The fatty liver index (FLI) and the NAFLD-liver fat score (NAFLD-LFS) are non-invasive liver fat indices introduced in clinical practice aiming for the early detection of NAFLD and are best validated for purposes in cross-sectional studies. However, their longitudinal accuracy has been proven to be diet-specific: changes in liver fat, obtained by a low-fat diet intervention, moderately correlate to changes in FLI and NAFLD-LFS, but this does not happen when undertaking a low-carbohydrate diet approach [96]. Additionally, one 6-month clinical trial on NAFLD patients undertaking a Mediterranean diet has shown significant improvements in FLI and NAFLD-LFS [95]. The impact of lifestyle interventions on fibrosis is less well-established, although long-term histologic evaluation has provided evidence of fibrosis amelioration [11]. In one cross-sectional study conducted on biopsy-proven NAFLD patients with T2DM, liver fibrosis was inversely associated with adherence to the Mediterranean diet at multivariate analysis, including multiple putative factors (age, gender, BMI, glycated hemoglobin) [122]. In addition, one prospective observational Greek study reported the same inverse association when evaluating liver fibrosis with non-invasive scores (namely, fibrosis-4 score (FIB-4), AST-to-platelet ratio (APRI) and BARD (BMI, AST/ALT Ratio, Diabetes) index) through cross-sectional analysis [123]. An inverse association between the Mediterranean diet and liver fibrosis also emerged through elastography after 6 months of intervention, suggesting a possible role for liver stiffness, together with CAP, in the assessment of treatment progress [47].

In conclusion, the nutritional landscape of interventional approaches for NAFLD treatment is burdened by the heterogeneity in study populations, duration of treatments, and the diversity of selected endpoints. The lack of longitudinal studies in nutritional patterns on the natural history of NAFLD makes it hard to assess strong benefits of either nutrients or dietary approaches, of which evidence relies on surrogate markers or short-term evaluation. The complex picture of the metabolic syndrome requires a careful, multidisciplinary method, because multiple causal agents impact on liver disease, and systemic morbidities are in return worsened by liver damage. The relative strong evidence of beneficial effects of the Mediterranean diet is counterbalanced by a lack of studies in other dietary models, of which usefulness is crucial for populations where the Mediterranean diet would be undertaken with difficultly, due to geographical and cultural discrepancies in food accessibility.

In addition, evidence of the harmful effect on liver fat content of late eating, rushed eating and attitude to snacking has highlighted the importance of behavioral aspects in nutrition, which are often underreported and thus require proper investigation. It is likely than more than one approach would ameliorate liver and systemic metabolism, and good communication with patients with respect to harmful nutrients, time of meals, and proper behavior might produce better results per se, as well as increase compliance and understanding. Aiming for improvements in insulin sensitivity can be the fil rouge of different approaches, which would not potentially be limited to restrictions of food, but also to eating times. In fact, the positive results on insulin sensitivity coming from the novel approaches of intermittent feeding have corroborated the concept of quality and timing of eating, with important implications in clinical management. The need for associating one nutritional intervention with inexpensive, non-invasive scores of either steatosis or fibrosis is essential in clinical settings, in order to assess benefits or futility. These algorithms should be included in future studies, with particular regard to liver fibrosis, which is the hardest prognostic factor in the setting of chronic liver disease.

## Figures and Tables

**Figure 1 nutrients-13-01977-f001:**
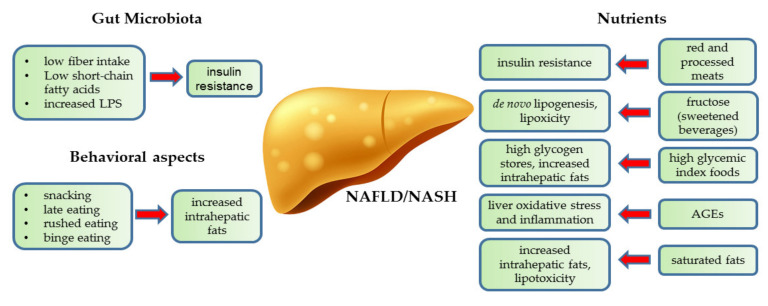
Impact of gut microbiota, behavioral aspects and specific nutrients on features of liver damage leading to Non Alcoholic Fatty Liver Disease/Non Alcoholic Steatohepatitis (NAFLD/NASH). Abbreviations: AGEs, advanced glycation end-products; LPS, lipopolysaccharide.

**Table 1 nutrients-13-01977-t001:** Clinical studies evaluating the impact of different dietary pattern on both hepatic and metabolic outcomes.

Type of Intervention	Study	Year	Design	Population	Duration of the Study	Staging and Grading (Liver Phenotype)	Results	Other Outcomes
Diet and physical activity								
Low-calorie, low-fat	Vilar-Gomez et al. [9].	2015	Observational study	293 White American obese/overweight biopsied NASH	52 weeks	Histology	25% resolution of NASH (*p* < 0.01) and 19% regression of fibrosis (*p* < 0.01)	Improvement in NAS associated with weight loss > 5% (*p* < 0.001)
Low-fat, low glycemic index	Wong et al. [93].	2013	RCT	154 Asian MRS-proven NAFLD	48 weeks	Radiology (MRS)	Mean change in hepatic fat of 6.7% intervention versus 2.1% control (*p* < 0.001)	Median weight loss of 5.6 kg intervention versus 0.6 kg control (*p* < 0.001)
Low glycemic index Mediterranean diet	Franco et al. [46].	2020	RCT	144 Caucasian CAP-based NAFLD	12 weeks	Non-invasive assessment of steatosis (CAP)	Median reduction of 61 points in CAP (*p* < 0.0001)	Reduction in HOMA-IR (not statistically significant)
Low-calorie	Promrat et al. [10].	2010	RCT	31 White American obese/overweight biopsied NASH	48 weeks	Histology	72% improvement in NAS score (*p* = 0.03)	Improvement in NAS associated with weight loss ≥ 7% (*p* < 0.001)
Mediterranean diet								
	Katsagoni et al. [47].	2018	RCT	63 Caucasian obese/overweight U.S.-based NAFLD	24 weeks	Non-invasive assessment of fibrosis	Reduction in liver stiffness, not statistically significant	Median weight loss of 13.7 kg (*p* < 0.05)
	Properzi et al. [94].	2018	RCT	56 Australian MRS-proven NAFLD	12 weeks	Radiology (MRS)	Mean change in hepatic fat of 32.4% intervention versus 25% control (low-fat diet) (*p* = 0.32)	0.2% reduction in HbA1c in the intervention arm (*p* < 0.045)
	Gelli et al. [95].	2017	Observational study	46 Caucasian U.S.-based NAFLD	24 weeks	Radiology (US), liver function tests, non-invasive assessment of steatosis (FLI, NAFLD-LFS)	20% of steatosis regression at US, reduction in ALT (*p* < 0.01), mean reduction of 12.7 points in FLI (*p* < 0.01), reduction of 0.7 points in NAFLD-LFS (*p* < 0.01)	Reduction of 1.8 points of BMI (*p* < 0.01), improvement in HOMA-IR (*p* < 0.01)
	Ryan et al. [41].	2013	Randomized cross-over trial	12 Australian obese non-diabetic biopsied NAFLD	6 weeks	Radiology (MRS)	Mean change in hepatic fat of 39% intervention (*p* = 0.012)	Improvement in insulin sensitivity by euglycemic clamp (*p* = 0.03)
Single nutrient evaluation								
Fiber supplementation	Stachowska et al. [89].	2020	Meta-analysis of 6 RCTs	242 Caucasian U.S.-based NAFLD	12 weeks	Liver function tests	Reduction in ALT (*p* = 0.001)	Reduction in 0.5 points of BMI (*p* = 0.009), improvement in HOMA-IR (*p* = 0.003)
Omega-3 supplementation (fish oil)	Qin et al. [84].	2015	RCT	80 Asian U.S.-based NAFLD	48 weeks	Liver function tests	Reduction in ALT (*p* < 0.05)	
Omega-3 supplementation (flaxseed, fish oil)	Nogueira et al. [81].	2016	RCT	60 Latin American biopsied NASH	24 weeks	Histology	Improvement in lobular inflammation (*p* = 0.05)	
Low-fructose, low glycemic index	Mager et al. [42].	2015	Observational study	26 White American children/adolescent biopsied NAFLD	24 weeks	Liver function tests	Reduction in ALT (*p* = 0.004)	Improvement in systolic blood pressure (*p* = 0.01), HOMA-IR (*p* = 0.03)
Omega-3 supplementation (olive oil)	Sofi et al. [82].	2010	Observational study	11 Caucasian U.S.-based NAFLD	48 weeks	Radiology (US), liver function tests	0.04-point reduction in Doppler Perfusion Index (*p* < 0.05), improvement in ALT (*p* = 0.04)	
Low-calorie diet								
Low-fat	Kabisch et al. [96].	2018	RCT (low-fat versus low-carbohydrate)	140 Caucasian pre-diabetic MRS-proven NAFLD	48 weeks	Radiology (MRS)	11% versus 10.2% mean change in hepatic fat (*p* = 0.59)	
DASH diet ^1^	Razavi et al. [97].	2016	RCT	60 Caucasian obese/overweight U.S.-based NAFLD	8 weeks	Radiology (US), liver function tests	Decreasing of steatosis grade in 80% of patients (*p* = 0.003), reduction in ALT (*p* =0.02)	Mean weight loss of 3.8 kg (*p* = 0.006), Improvement in HOMA-IR (*p* = 0.01) and CRP (*p* = 0.03)
Very low-calorie diet (800 kcal/day)	Scragg et al. [98].	2020	Observational study	45 Caucasian U.S.-based NAFLD	8 weeks	Non-invasive assessment of fibrosis	Mean reduction of 5.1 points at transient elastography (*p* = 0.001)	Mean weight loss of 10.3 kg (34% of patients reached > 10% weight loss), improvement in HOMA-IR
Low-carbohydrate	Sevastianova et al. [22].	2011	Observational study	8 Caucasian homozygous rs738409 PNPLA3 G allele versus 10 homozygous rs738409 PNPLA3 C allele MRS-proven NAFLD carriers	6 days	Radiology (MRS)	Reduction in liver fat of 45% (*p* < 0.001) in G allele carriers versus 18% in C allele carriers (*p* < 0.01)	
Isocaloric diet								
Low-carbohydrate, high protein	Mardinoglu et al. [99].	2018	Observational study	10 Caucasian obese MRS-proven NAFLD	2 weeks	Radiology (MRS)	Mean reduction in liver fat of 43.8% (*p* = 0.027)	
High protein	Markova et al. [100].	2017	RCT	37 Caucasian diabetic MRS-proven NAFLD	6 weeks	Radiology (MRS)	Reduction in liver fat of 36–48% (*p* = 0.0002)	Improvement in insulin sensitivity by euglycemic clamp
Low-fat, low glycemic index	Utzschneider et al. [101].	2013	RCT	35 White American MRS-proven NAFLD	4 weeks	Radiology (MRS)	Mean reduction in liver fat of 2.2% (*p* = 0.002)	Improvement insulin sensitivity by Matsuda index (*p* < 0.05)

^1^ Low-calorie, low-fat diet, restricted in refined grains and rich in fruits/vegetables. Abbreviations: ALT, alanine aminotransferase; BMI: Body Mass Index; CAP, controlled attenuation parameter; CRP, C-reactive protein; DASH, Dietary Approaches to Stop Hypertension; FLI, Fatty Liver Index; HOMA-IR, Homeostatic Model Assessment of Insulin Resistance; IL-6, interleukin-6; MRS, magnetic resonance spectroscopy; NAFLD, non-alcoholic fatty liver disease; NAFLD-LFS: NAFLD-liver fat score; NAS, NAFLD activity score; NASH, non-alcoholic steatohepatitis; PNPLA3, Patatin-like phospholipase domain-containing 3; RCT, randomized controlled trial; TNF-α, tumor necrosis factor-α; US, ultrasound.

## Data Availability

All data are publicly available.

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
