# Peer review of "Beyond the Paradigm of Weight Loss in Non-Alcoholic Fatty Liver Disease: From Pathophysiology to Novel Dietary Approaches"

_nutrients, 2021, doi:10.3390/nu13061977_

Round 1

Reviewer 1 Report

In this paper, the nutritional impact to NAFLD/NASH is extensively overviewed. There are a lot of controversial reports in this field, but generally the authors seemed to have done a good job. I would like to raise a couple of points.

1) In most part of the paper, NAFLD and NASH are discussed together. Since NASH is more difficult to be resolved than NAFL without inflammatory changes, and it is almost impossible to diagnose each stage of NASH (especially fibrosis) without biopsy, this point should be taken with care. In Table1, the assessment of the effects on the development of NAFLD/NASH is variable among the studies, and thus the conclusions may not be equally convincing. I suggest this point should be mentioned in an earlier chapter in order to avoid misunderstanding in the later parts.

2) Since the paper only focused on dietary intervention, I am afraid the title seemed misleading. Although I understand pharmacotherapy is out of scope of this paper, it would be nice for the authors to briefly comment on some drugs which exert body weight lowering effects and are now used to tackle with NAFLD/NASH (i.e., GLP1RA, or SGLT2 inhibitors). This would strengthen the messages of this paper, if they aimed to discuss the significance of ‘effects beyond body weight loss’.

3)  One of the most complicated issues about diet therapy is ethnicity-specific outcome. This is important since the prevalence of NAFLD/NASH is quite different among populations, and Asians are more prone to NAFLD/NASH without significant obesity. There must be several reasons for this, such as genetic backgrounds or general life styles. The microbiome is also significantly diverse among different ethnic groups. Therefore, it would be nice to discuss these points in a small but independent chapter. The authors should make clear which ethnic groups were included in the clinical trials listed in Table 1.

4) I just wonder how much of molecular mechanisms of dietary intervention has been investigated either in humans or mouse models. There are several key components and potential drug targets of NAFLD/NASH, such as PPARs and FXR. Are the expressions of these genes affected by the diet therapy?

Author Response

Reviewer 1 

In this paper, the nutritional impact to NAFLD/NASH is extensively overviewed. There are a lot of controversial reports in this field, but generally the authors seemed to have done a good job. I would like to raise a couple of points. 

1) In most part of the paper, NAFLD and NASH are discussed together. Since NASH is more difficult to be resolved than NAFL without inflammatory changes, and it is almost impossible to diagnose each stage of NASH (especially fibrosis) without biopsy, this point should be taken with care. In Table1, the assessment of the effects on the development of NAFLD/NASH is variable among the studies, and thus the conclusions may not be equally convincing. I suggest this point should be mentioned in an earlier chapter in order to avoid misunderstanding in the later parts. 

- We thank reviewer 1 for his suggestion. We have added the required explanations in the early part of the review (lines 44-51).  

2) Since the paper only focused on dietary intervention, I am afraid the title seemed misleading. Although I understand pharmacotherapy is out of scope of this paper, it would be nice for the authors to briefly comment on some drugs which exert body weight lowering effects and are now used to tackle with NAFLD/NASH (i.e., GLP1RA, or SGLT2 inhibitors). This would strengthen the messages of this paper, if they aimed to discuss the significance of ‘effects beyond body weight loss’. 

- We agree and have edited the title to stress the focus of the review on nutritional interventions. Also, a small section on pharmacological options was added to allow to relate to the effects that can be potentially be achieved by drugs. 

3) One of the most complicated issues about diet therapy is ethnicity-specific outcome. This is important since the prevalence of NAFLD/NASH is quite different among populations, and Asians are more prone to NAFLD/NASH without significant obesity. There must be several reasons for this, such as genetic backgrounds or general life styles. The microbiome is also significantly diverse among different ethnic groups. Therefore, it would be nice to discuss these points in a small but independent chapter. The authors should make clear which ethnic groups were included in the clinical trials listed in Table 1. 

- This is an important comment and we have expanded to include the available evidence of nutritional interventions in the context of genetic susceptibility. Overall, there is only scarce data of the impact of genetic background and environment with regards to the response in lifestyle interventions in NAFLD. In addition, we expanded table 1 and included the populations and background when available. 

4) I just wonder how much of molecular mechanisms of dietary intervention has been investigated either in humans or mouse models. There are several key components and potential drug targets of NAFLD/NASH, such as PPARs and FXR. Are the expressions of these genes affected by the diet therapy? 

- We welcome this comment and have added a short paragraph (lines 132-138) detailing some (as much as possible within the scope of this review) mechanistic studies exploring nutrient-related expression of key genes in liver metabolism.  

Reviewer 2 Report

In addition to weight loss, which has been recognized as the best method for the prevention and treatment of NAFLD, the author approached and organized various aspects of the treatment methods suitable for individuals. I think this attempt is a very important task that goes beyond inducing interest.

However, this paper contains too much explanation, which hinders understanding what the author was trying to say. In order to communicate the intentions of this paper and help readers understand it, important contents need to be consistently listed and organized, and the duplicated contents and unnecessary parts need to be deleted.

  1. The author said in the abstract that the improvement of insulin sensitivity is essential, but there is not enough content in the text to support this. In particular, "2. Improving insulin resistance as metabolic endpoint for lifestyle intervention" should provide a deeper explanation and rationale for the causality of insulin resistance and NAFLD.
  2. In conjunction with the above questions, explain the relationship between life style and insulin resistance. List the relationship between nutrients, such as fructose, lipids, fiber of "5. Diverse impact of nutrients on NAFLD" in the main text and insulin resistance and NAFLD. In addition, “3. "Response to lifestyle intervention between genes and environment" interferes with the flow of the article, so changing the order seems essential.
  3. In the text, it is emphasized that the Mediterranean diet is important for improving NAFLD. Is this also related to insulin resistance?
  4. Conclusion of “Aiming at improvement in insulin sensitivity can be the fil rouge of different approaches, that would not potentially be limited to restriction of food, but also to eating time.” This sentence is confusing the readers. It would be better to provide additional explanations or comments following that sentence.
  5. Lines 80 to 89 on page 2 overlap with the contents of the introduction, and there seems to be little relevance between the subtitle and the contents.

Author Response

Reviewer 2 

In addition to weight loss, which has been recognized as the best method for the prevention and treatment of NAFLD, the author approached and organized various aspects of the treatment methods suitable for individuals. I think this attempt is a very important task that goes beyond inducing interest. 

However, this paper contains too much explanation, which hinders understanding what the author was trying to say. In order to communicate the intentions of this paper and help readers understand it, important contents need to be consistently listed and organized, and the duplicated contents and unnecessary parts need to be deleted. 

- In the revision we have highlighted the central message and removed redundancy. 

1) The author said in the abstract that the improvement of insulin sensitivity is essential, but there is not enough content in the text to support this. In particular, "2. Improving insulin resistance as metabolic endpoint for lifestyle intervention" should provide a deeper explanation and rationale for the causality of insulin resistance and NAFLD. 

- This is an important comment. We expanded section 2 on page 3 focusing on pre-clinical studies connecting insulin resistance and NAFLD mechanistically. In addition, we added evidence from lifestyle intervention studies highlighting the beneficial effect of improvement of insulin sensitivity and liver surrogates. This translational data now positions the central message of the review around the clinical  trials.  

2) In conjunction with the above questions, explain the relationship between life style and insulin resistance. List the relationship between nutrients, such as fructose, lipids, fiber of "5. Diverse impact of nutrients on NAFLD" in the main text and insulin resistance and NAFLD. In addition, “3. "Response to lifestyle intervention between genes and environment" interferes with the flow of the article, so changing the order seems essential. 

  • As detailed above, this part was expanded. Now section 5 (page 9) summarizes relevant nutrients and their putative role in both insulin resistance and NAFLD.  Also, we rearranged the order within section 3 (page 4) to improve the flow of reasoning.  

3) In the text, it is emphasized that the Mediterranean diet is important for improving NAFLD. Is this also related to insulin resistance? 

- The review includes evidence on the beneficial effects of the Mediterranean type of diets on both NAFLD and insulin resistance. We added sentences (lines 347-350) to clarify and strengthen the association according to the reviewer’s suggestion. 

4) Conclusion of “Aiming at improvement in insulin sensitivity can be the fil rouge of different approaches, that would not potentially be limited to restriction of food, but also to eating time.” This sentence is confusing the readers. It would be better to provide additional explanations or comments following that sentence. 

- We reworded and clarified to support the potential benefits of intermittent fasting, which the misleading sentence referred to initially. 

5) Lines 80 to 89 on page 2 overlap with the contents of the introduction, and there seems to be little relevance between the subtitle and the contents. 

- We have deleted the overlapping lines. 

Reviewer 3 Report

This review is timely and addresses important points that are typically overlooked or ignored when treatment approaches for NAFLD/NASH are conventionally discussed. The overview of studies listed in the tables is helpful and of practical value for other researchers. The paragraphs on e.g. consequences of behavior; different isocaloric dietary regimens etc are refreshing because they deviate from conventional reviews. I do support publication of this review and would like to ask the authors to have a closer look at the following points which require some adjustment of language, more nuance, mechanistic detail or simply correction:.

  1. In the Introduction, the authors cited their own study (ref 7) with the statement that most NAFLD patients are overweight or obese and suffer from T2M.  Although many patients with NAFL (and thus NAFLD) are indeed fulfilling these criteria, a considerable portion of patients, in particular those with severe stages of NASH, are not necessarily overweight/obese. Although their sentence is not incorrect, inexpert readers of this Journal may get the impression that lean patients could be neglected which is not the case at all.
  2. In the second paragraph the authors discuss the role of IR in the etiology of disease. It may be emphasized here that the group of 'obese' patients is very heterogenous (e.g. metabolically healthy obese vs obese subjects with only peripheral or only central IR), as the NAFLD/NASH patient group. The diversity is in part due to the different stages of dysmetabolism but also a consequence of which peripheral organ is impacted most (think of muscle-IR versus WAT-IR). The heterogenous nature of NAFLD/NASH should be discussed in more detail (in one of the first paragraphs) so that the reader understands why nutritional approaches do not result in uniform effects across a population.
  3. line 171: this is a very vague statement ('alterations' is practically everything that changes up or down and not very meaningful). The integrity of the gut and the functionality of both the epithelial cell layer as well as the nutritional (e.g. lipid and amino acid uptake) processes in the jejunum are of course always microbiota-related. I would not consider this as a microbiota-related damage because it is a disturbance of the metabolic and inflammatory homeostasis of the cells of the gut, that can (but does not have to) affect its barrier function. The microbiota could better be discussed as an integral part of body homeostasis rather than as an entity with independent action. 
  4. The authors discussed the effects of Mediterr. diets and cite several of the reported benefit. In is important to emphasize somewhere in the review that the entire dietary pattern rather than single constituents or specific fatty acids (e.g. total PUFAs or specific PUFAs) are critical. It is equally important to emphasize that excess intake of any type of fatty acid, even the most antiinflammatory PUFAs, will cause hypertrophy of cells in target organs (typically first in adipocytes and later in other organs such as hepatocytes or vascular cells).   Thus, the effects in line 220 onwards are effects of complex foods (and the eating pattern in specific regions of the world) rather than its constituents.
  5. The behavioral aspects discussed in 4.2 constitute an important element of this review. A reference to 4.2 should also be part of the concluding remarks at the end of the review.
  6. Figure 1: low microbial richness is a not defined term and microbiota diversity can be calculated in many ways, and depends on the methodologies used for sequencing and post-sequencing/data analysis. Dysbiosis or functional impairment of microbiota may be more appropriate expressions, when defined and explained with examples. For instances, alterations in bile acid processing, impaired capacity to process complex carbohydrates and complex proteins.  
  7. Line 320: why could? A high glycemic index food pattern is unfavorable in diabetes.
  8. The authors should provide more mechanistic detail in the paragraph on oxidative stress. This is the only paragraph which requires more extensive revision. When ATP is generated and the complexes of the electron transfer chain (ETC) are operational, a certain number of electrons are 'lost' (this is a stochastic process) and these electrons contribute to the formation of ROS which react with many intracellular constituents or lipids in the mitochondrial membrane. One of the endogenously formed reactive molecules is for instance 4-HNE. The higher the flow of electrons through the complexes of the ETC, the more electrons are potentially lost. Thus high metabolic rate will intrinsically cause more losses and thus also more damage. However, the latter depends on the antioxidant capacity of the cell in question. Typically, the enzymes of the antioxidant defense mechanisms are first increased in the etiology of many metabolic overload conditions and, at later stage, they numbers decline and a vicious cycle starts. The present paragraph requires in my view more detail on the molecular processes of oxidative stress in the context of NAFLD/NASH and obesity and how weight loss may affect these processes.
  9. line 356: the authors discussed the role of ceramides in this paragraph. When it comes to liver lipids, diacylglycerides (DAGs) appear to be critical for insulin sensitivity of the organ and the development of a proinflammatory milieu. The role of DAGs should be discussed (e.g. see review by Petersen and Shulman Trends Pharmacol Sci 2017)
  10. There is an extremely high variation between subjects regarding their glycemic response as shown by Zeevi et al in Cell 2015). There remarkable differences between human responses to the same foods may account for many of the apparent weak or modest effects of dietary treatments. It could increase the awareness of readers that it is unlikely that a particular dietary regimen X will have an uniform effect and may explain the apparent discrepancies observed with nutritional lifestyle approaches.

Author Response

Reviewer 3 

This review is timely and addresses important points that are typically overlooked or ignored when treatment approaches for NAFLD/NASH are conventionally discussed. The overview of studies listed in the tables is helpful and of practical value for other researchers. The paragraphs on e.g. consequences of behavior; different isocaloric dietary regimens etc are refreshing because they deviate from conventional reviews. I do support publication of this review and would like to ask the authors to have a closer look at the following points which require some adjustment of language, more nuance, mechanistic detail or simply correction: 

  • We thank the reviewer for these positive remarks. 

In the Introduction, the authors cited their own study (ref 7) with the statement that most NAFLD patients are overweight or obese and suffer from T2M.  Although many patients with NAFL (and thus NAFLD) are indeed fulfilling these criteria, a considerable portion of patients, in particular those with severe stages of NASH, are not necessarily overweight/obese. Although their sentence is not incorrect, inexpert readers of this Journal may get the impression that lean patients could be neglected which is not the case at all. 

  • We thank the reviewer for this relevant point and added the aspect of lean NAFLD in this paragraph. 

In the second paragraph the authors discuss the role of IR in the etiology of disease. It may be emphasized here that the group of 'obese' patients is very heterogenous (e.g. metabolically healthy obese vs obese subjects with only peripheral or only central IR), as the NAFLD/NASH patient group. The diversity is in part due to the different stages of dysmetabolism but also a consequence of which peripheral organ is impacted most (think of muscle-IR versus WAT-IR). The heterogenous nature of NAFLD/NASH should be discussed in more detail (in one of the first paragraphs) so that the reader understands why nutritional approaches do not result in uniform effects across a population. 

  • The revision now includes details on the diverse phenotypes, particularly detailing IR in different compartments. 

line 171: this is a very vague statement ('alterations' is practically everything that changes up or down and not very meaningful). The integrity of the gut and the functionality of both the epithelial cell layer as well as the nutritional (e.g. lipid and amino acid uptake) processes in the jejunum are of course always microbiota-related. I would not consider this as a microbiota-related damage because it is a disturbance of the metabolic and inflammatory homeostasis of the cells of the gut, that can (but does not have to) affect its barrier function. The microbiota could better be discussed as an integral part of body homeostasis rather than as an entity with independent action.  

  • We agree and have removed the statement on alterations and edited this text passage. In particular, we removed parts that could have suggested a “straight froward” interaction of microbiota and liver damage. The multifaceted nature of the diverse interactions in this area – involving immune and endothelial cells on top of parenchymal cells, blood flow and nutrients is beyond this review’s content.   

The authors discussed the effects of Mediterr. diets and cite several of the reported benefit. In is important to emphasize somewhere in the review that the entire dietary pattern rather than single constituents or specific fatty acids (e.g. total PUFAs or specific PUFAs) are critical. It is equally important to emphasize that excess intake of any type of fatty acid, even the most antiinflammatory PUFAs, will cause hypertrophy of cells in target organs (typically first in adipocytes and later in other organs such as hepatocytes or vascular cells).   Thus, the effects in line 220 onwards are effects of complex foods (and the eating pattern in specific regions of the world) rather than its constituents. 

  • We have added this important integration in the review. 

The behavioral aspects discussed in 4.2 constitute an important element of this review. A reference to 4.2 should also be part of the concluding remarks at the end of the review. 

  • Thank you for this suggestion. We added accordingly. 

Figure 1: low microbial richness is a not defined term and microbiota diversity can be calculated in many ways, and depends on the methodologies used for sequencing and post-sequencing/data analysis. Dysbiosis or functional impairment of microbiota may be more appropriate expressions, when defined and explained with examples. For instances, alterations in bile acid processing, impaired capacity to process complex carbohydrates and complex proteins.   

  • We agree and have modified Figure 1 accordingly. 

  1. Line 320: why could? A high glycemic index food pattern is unfavorable in diabetes. 

  • We have modified the text accordingly. 

The authors should provide more mechanistic detail in the paragraph on oxidative stress. This is the only paragraph which requires more extensive revision. When ATP is generated and the complexes of the electron transfer chain (ETC) are operational, a certain number of electrons are 'lost' (this is a stochastic process) and these electrons contribute to the formation of ROS which react with many intracellular constituents or lipids in the mitochondrial membrane. One of the endogenously formed reactive molecules is for instance 4-HNE. The higher the flow of electrons through the complexes of the ETC, the more electrons are potentially lost. Thus high metabolic rate will intrinsically cause more losses and thus also more damage. However, the latter depends on the antioxidant capacity of the cell in question. Typically, the enzymes of the antioxidant defense mechanisms are first increased in the etiology of many metabolic overload conditions and, at later stage, they numbers decline and a vicious cycle starts. The present paragraph requires in my view more detail on the molecular processes of oxidative stress in the context of NAFLD/NASH and obesity and how weight loss may affect these processes. 

  • Another important comment in this context. While in-depth details of the oxidative injury and ETC are beyond the scope of this review, we amended and added some mechanistic details derived from both murine and human studies and the effects of nutrition and weight loss on ox stress in NAFLD. 

line 356: the authors discussed the role of ceramides in this paragraph. When it comes to liver lipids, diacylglycerides (DAGs) appear to be critical for insulin sensitivity of the organ and the development of a proinflammatory milieu. The role of DAGs should be discussed (e.g. see review by Petersen and Shulman Trends Pharmacol Sci 2017) 

  • The reference and implications were added.  

There is an extremely high variation between subjects regarding their glycemic response as shown by Zeevi et al in Cell 2015). There remarkable differences between human responses to the same foods may account for many of the apparent weak or modest effects of dietary treatments. It could increase the awareness of readers that it is unlikely that a particular dietary regimen X will have an uniform effect and may explain the apparent discrepancies observed with nutritional lifestyle approaches. 

  • This constitutes an important aspect and we added this relevant point (lines 706-711). 

Round 2

Reviewer 1 Report

My points were well taken, and the manuscript has been improved significantly.